# Advanced Glycation End Products (AGEs) and Chronic Kidney Disease: Does the Modern Diet AGE the Kidney?

**DOI:** 10.3390/nu14132675

**Published:** 2022-06-28

**Authors:** Amelia K. Fotheringham, Linda A. Gallo, Danielle J. Borg, Josephine M. Forbes

**Affiliations:** 1Glycation and Diabetes Complications, Mater Research Institute, Translational Research Institute, The University of Queensland, Brisbane, QLD 4102, Australia; amelia.fotheringham@mater.uq.edu.au; 2Faculty of Medicine, University of Queensland, Brisbane, QLD 4072, Australia; danielle.borg@mater.uq.edu.au; 3School of Health and Behavioural Science, University of the Sunshine Coast, Sippy Downs, QLD 4556, Australia; lgallo@usc.edu.au; 4Pregnancy and Development, Mater Research Institute, The University of Queensland, Brisbane, QLD 4101, Australia

**Keywords:** advanced glycation end products (AGEs), chronic kidney disease (CKD), diabetic kidney disease (DKD), diet, diabetes, ultra-processed foods, receptor for advanced glycation end products (RAGE)

## Abstract

Since the 1980s, chronic kidney disease (CKD) affecting all ages has increased by almost 25%. This increase may be partially attributable to lifestyle changes and increased global consumption of a “western” diet, which is typically energy dense, low in fruits and vegetables, and high in animal protein and ultra-processed foods. These modern food trends have led to an increase in the consumption of advanced glycation end products (AGEs) in conjunction with increased metabolic dysfunction, obesity and diabetes, which facilitates production of endogenous AGEs within the body. When in excess, AGEs can be pathological via both receptor-mediated and non-receptor-mediated pathways. The kidney, as a major site for AGE clearance, is particularly vulnerable to AGE-mediated damage and increases in circulating AGEs align with risk of CKD and all-cause mortality. Furthermore, individuals with significant loss of renal function show increased AGE burden, particularly with uraemia, and there is some evidence that AGE lowering via diet or pharmacological inhibition may be beneficial for CKD. This review discusses the pathways that drive AGE formation and regulation within the body. This includes AGE receptor interactions and pathways of AGE-mediated pathology with a focus on the contribution of diet on endogenous AGE production and dietary AGE consumption to these processes. We then analyse the contribution of AGEs to kidney disease, the evidence for dietary AGEs and endogenously produced AGEs in driving pathogenesis in diabetic and non-diabetic kidney disease and the potential for AGE targeted therapies in kidney disease.

## 1. The Western Diet as a Risk Factor for Kidney Disease

Chronic kidney disease (CKD) is the 12th global cause of mortality with all age mortality attributable to CKD increasing by 41.5% since the 1980s [1]. Furthermore, patients with CKD have been amongst the most vulnerable to the recent COVID19 pandemic [2] with the risk of COVID19 mortality increasing nearly 4-fold in kidney transplant recipients or those receiving dialysis [3]. With the pathogenesis of CKD often likened to accelerated kidney aging, epidemiological evidence suggests that underlying causes of CKD have shifted in the last 50–100 years from glomerulonephritis and congenital diseases towards diabetes and hypertension [4]. Type 2 diabetes (T2D) alone accounts for more than 40% of new CKD cases [1] and more than 50% of patients entering renal replacement therapy [4]. The modern western diet, high in animal protein, saturated- and trans-fats, sugar and salt, whilst low in fruits, vegetables, fibre and other essential nutrients, has been implicated in this rising incidence of CKD, as a result of direct effects on the kidneys and rising rates of obesity, hypertension and type 2 diabetes [5,6,7].

One major factor that has changed in the diet over this period of rising CKD incidence is the increased consumption of processed and ultra-processed foods [8], which are also linked to increased risk of cancer [9], T2D [10], cardiovascular disease (CVD) [11,12] and all-cause mortality [11,13,14], and ultra-processed food consumption appears to have risen during the COVID19 pandemic and associated lockdowns [15,16]. Modern food production, including the use of high temperatures, high pressure, dehydration, decompression, irradiation, salt, and preservatives to extend shelf life and palatability, significantly alters proteins and lipids, forming post-translational modifications, including advanced glycation end products (AGEs) within foods [17,18]. As the kidney plays a major role in the clearance of AGEs from the body, there has been much debate as to whether consumption of dietary AGEs can precipitate and/or contribute to progression of CKD and, hence, whether lowering dietary AGE intake or targeting this pathway might be of therapeutic benefit.

Beyond dietary AGEs, increased consumption of energy dense, nutrient poor foods, which are high in sugars and salt, and low in essential nutrients, actively contribute to hemodynamic and metabolic abnormalities, culminating in hypertension, obesity, and T2D which, in turn, facilitate endogenous AGE production [19]. Circulating concentrations of AGEs are positively associated with diabetic kidney disease (DKD) [20], loss of renal function in diabetes [21], and all cause and CVD mortality [22,23,24,25], with glycation considered one of the major pathways to end organ complications in diabetes [19].

## 2. AGE Chemistry

AGEs can be produced via several chemical pathways, with the Maillard reaction being the most well-described, non-enzymatic chemistry. This reaction, named after the French physician and chemist responsible for its discovery, is the covalent non-enzymatic attachment of a reducing sugar to an amine or amino acid (AA) and can proceed via a number of chemical pathways. The carbonyl groups in reducing sugars such as glucose, fructose, ribose and mannose are, by nature, reactive towards amine groups, and this chemical attraction underpins Maillard chemistry [26], forming a Schiff’s base (Figure 1 green). These then undergo further rearrangement to produce the Amadori product or a Heyn’s product, depending on whether the originating sugar contained an aldehyde or ketone group [27]. The resulting Amadori products combine with amine groups on proteins to form AGEs or undergo further degradation into reactive carbonyls (Figure 1). These highly reactive carbonyls also react with amino groups of free amines, peptide or proteins to form AGEs [27] (Figure 1 purple). AGE formation is enhanced by factors including heat, the presence of oxidants, increases in pH, atmospheric pressure and higher concentrations of reducing sugars and AAs. In the early steps of this reaction, the reactions are reversible, but become permanent modifications once AGEs are formed [27].

Maillard chemistry is incredibly diverse. This complexity is in part explained by the many divergent changes to the sugar moiety. For example, sugars may undergo various reactions such as oxidation, dehydration and fragmentation reactions prior to the attachment to amines [28]. Important reactive intermediates of the Maillard reaction are the reactive carbonyls, including, but not limited to, 3-deoxyglucose (3-DG), glyoxal and methylglyoxal (MGO) [29,30,31]. The accumulation of such molecules is known as dicarbonyl stress, and can dramatically accelerate the formation of AGEs, as they are up to 20,000 times more potent glycating agents than glucose [28]. Due to the complexity and variety of starting components, the term AGE refers to a very large heterogeneous population of variable size and structure (Figure 1). These range from low molecular weight, single AGE-modified amino acids to complex high molecular weight AGEs with AGE-crosslinks such as glyoxal lysine dimer (GOLD), 3-deoxyglucosone lysine dimer (DOLD).

## 3. Factors Regulating AGE Accumulation and Turnover in the Body

In humans and animals, the whole-body AGE load represents the balance between endogenous AGE production, exogenous AGE absorption, and AGE clearance mechanisms.

### 3.1. Endogenous AGE Production—Beyond the Maillard Reaction

In addition to the Maillard reaction, there are other well-characterised pathways of endogenous AGE production within the body. These include glucose auto-oxidation and lipid peroxidation, generating α-oxoaldehydes, which can subsequently react with monoacids to generate AGEs [32,33] (Figure 1). Additionally, the potent glycating agents and highly reactive dicarbonyls, MGO and 3-DG, are produced in cells from fatty acid catabolism and anaerobic glycolysis [34], and catabolism of ketone bodies [35] (Figure 1). Hence, in states of metabolic or oxidative stress, there is excessive generation of endogenous AGEs.

In the biomedical field, the most commonly measured endogenous AGEs include, carboxymethyl-lysine (CML) and carboxyethyl-lysine (CEL), and cross-linking dimers, such as pentosidine, glyoxal-lysine dimer (GOLD) and methylglyoxal-lysine dimer (MOLD) [36]. AGEs in the skin are also measured both directly by assessment of AGE-modified collagen in skin biopsies [37,38] but increasingly via skin auto-fluorescence, which is a non-invasive point-of-care measurement [39] and considered an indicator of systemic AGE load [40].

Under physiological conditions, AGE formation is most common on long-lived proteins in the circulation and connective tissues [19], such as structural components of cellular basement membranes and proteins including globulin, immunoglobulins and albumin [30,41]. However, conditions such as chronic inflammation, hyperlipidaemia and oxidative stress further accelerate the process of glycation, increasing the likelihood of short-lived proteins also becoming AGE modified [42]. In the context of diabetes and dysregulated glucose metabolism, AGEs such as glycated haemoglobin (HbA1C) and fructosamine-modified albumin are used to monitor long-term glucose control. While chronic conditions, such as diabetes, hypertension, and CKD, are associated with increased levels of circulating AGEs and accumulation of AGE modified proteins [43,44,45], little is understood about AGE homeostasis in healthy individuals and how this may be interrupted to contribute to disease risk.

### 3.2. Exogenous AGE Sources

AGEs and their highly reactive precursors are produced during food manufacturing. In particular, cooking and food processing conditions favour AGE production. Additionally, AGEs are widely used in the food industry to improve flavour, shelf-life, colour, aroma and texture [46]. High-temperature and dry-cooking methods (frying, baking and broiling) produce higher AGE concentrations than low temperature, aqueous cooking methods such as steaming and boiling [47,48]. Smoking is another exogenous source of AGEs [49]; however, this is outside the scope of this review.

#### 3.2.1. Quantifying AGEs in Commonly Consumed Foods

Several studies have aimed to estimate the AGE load found in commonly consumed foods, with the goal of generating reproducible databases for research purposes; however, quantifying AGEs in food has proven challenging. Studies using immunological methods [47,48,50] suggest that highly industrialised processes produce food with the highest AGE content, such as biscuits and cakes [50], and this is exacerbated for products high in saturated fat [47,48]. In the western diet, however, cooked meats likely constitute the greatest source of dietary AGEs due to both relative AGE content and high consumption rates [48]. More recently, these data have been questioned due to limitations with immunological methods, leading to the establishment of ultra-high performance liquid chromatography-MS/MS (UPLC MSMS)-derived databases of CML [51] and CML, CEL and MG-H1 [52] quantities in commonly consumed foods. Based on these more recent findings, foods such as peanut butter, manufactured biscuits and cakes, and canned and processed meats had the highest quantities of AGEs [52].

#### 3.2.2. Absorption of Dietary AGEs

There is much controversy regarding the quantity and processes used for AGE trafficking across the gastrointestinal (GI) tract. It remains to be fully understood whether dietary AGEs contribute to disease via: (i) direct entry into the circulation, resulting in tissue deposition, inflammation, and increased burden on clearance mechanisms; (ii) impacting intestinal health, permeability and entero-endocrine signalling; (iii) modulation of the GI tract microbiome, or; (iv) promotion of inflammation within the GI tract. It is possible that all these factors act in concert to promote the pathophysiology seen in the context of increased dietary AGE consumption.

The heterogeneity of AGEs that can be produced in food processing is much greater than those produced physiologically [53]. Therefore, well-characterised AGEs such as CML, CEL, pentosidine and pyralline are commonly used as biomarkers to gauge AGE load and uptake from the diet (Table 1). Generally, it is approximated from human and animal studies that between 10–30% of dietary AGEs are absorbed from the GI tract [54,55]. However, extensive analysis of the literature shows little consensus, with some studies suggesting greater uptake in humans [56], particularly in infants [57]. There have been many studies examining the uptake, kinetics, and bio-distribution of orally and intravenously administered AGEs and AGE precursors (Table 1). Despite this, there have been no technologies developed to accurately map this in humans in real-time. Such technologies would strengthen our understanding of the link between dietary AGEs and their associated pathologies.

### 3.3. AGE Clearance

Another major factor regulating AGE homeostasis is their clearance from the body. At the tissue level, this occurs through cellular proteolytic systems, which endocytose AGEs and break them down via receptor-mediated and non-receptor-mediated pathways into AGE peptides, which are then released back into the circulation [71]. At the systemic level, clearance of AGEs is thought to occur via the liver [72,73] and the kidney [63], where clearance of not only AGEs, but reactive carbonyl precursors, such as MG-H1, 3-DG and glyoxal and AGE-peptides, is important to maintain AGE homeostasis, although trafficking studies in humans are lacking. Animal models suggest that AGEs are filtered by the glomeruli, reabsorbed by proximal tubule cells and further processed and cleared into the urine [63,74]. In humans, AGE concentrations are commonly inversely related to renal function [42,44,75,76]. More recently, Haus et al. demonstrated that in obese but healthy humans, during a 24-h period of hyperglycaemia, plasma concentrations of several AGEs and oxidative products decreased concordant with an increase in the fractional excretion of these products into the urine [77]. Given its role in AGE clearance, the kidney has been highlighted as an important site of AGE mediated pathology [78,79,80,81,82,83,84], with renal function vital for AGE homeostasis, but potentially vulnerable to AGE-mediated damage.

Animal models demonstrate that uptake and clearance of intravenously administered AGEs also occur via the liver by endothelial, Kupffer and parenchymal cells [72]. However, the rate of clearance is dependent on the degree of AGE modification, with minimally AGE-modified bovine serum albumin (AGE-BSA) remaining in the circulation for significantly longer duration [73]. Concordant with the kidney, rodent models suggest that the liver also appears to be vulnerable to AGE-mediated pathology, particularly in the context of high AGE consumption [83,85].

## 4. AGE Receptors—Facilitators of Clearance and Mediators of Pathology

There are a number of receptors that have been characterised as binding AGEs, including the Receptor for Advanced Glycation End Products (RAGE), AGE Receptor 1 (AGER1; OST-48), AGE Receptor 2 (AGER2; 80K-H), AGE Receptor 3 (AGER3; Galectin 3) and the class A macrophage scavenger receptors types 1 and 2 [86]. The majority of studies focus on RAGE, since AGE binding induces cellular signal transduction. Other AGE receptors may play a role in AGE clearance and detoxification [87], such as AGER1, which has also garnered significant attention.

### 4.1. RAGE

RAGE, a member of the immunoglobulin superfamily of receptors [88,89], is the most widely studied of the AGE receptors. As a multi-ligand, pattern recognition receptor, RAGE also binds to s100 calgranulins [90], amphoterin/high mobility group box 1 protein (HMGB1) [91], lipopolysaccharide (LPS) [92], β amyloid [93], transthyretin [94], Mac-1 [95], complement 1q [96] and potentially DNA [92]. RAGE is highly expressed on mucous membranes such as those present in the lung and the GI tract, and within the immune system [97]. In the healthy kidney, RAGE localisation and gene expression occurs in the vascular smooth muscle, the epithelia of the proximal and distal tubule and is significantly upregulated in various inflammatory and non-inflammatory disease settings [98,99]. RAGE is significantly upregulated on podocytes in multiple diseases settings including diabetic nephropathy [99,100,101].

The RAGE gene, AGER, can be differentially spliced to form more than 20 different variants [102,103]. The full length form of RAGE (flRAGE) consists of three immunoglobulin-like domains known as the V1(variable), C1 and C2 domains; a transmembrane helix and a short highly charged cytoplasmic domain that is essential for signal transduction (Figure 2) [104]. As well as the full length isoform, there is an N terminal truncated form of the receptor that lacks the AGE binding capacity (Figure 2) [105]. Two secreted forms of RAGE exist, which form the circulating pool: endogenous secretory RAGE (esRAGE), which is a product of AGER transcription and soluble RAGE (sRAGE), which is cleaved from cell membranes by the proteases a-disintegrin and metalloproteinase domain-containing protein 10 (ADAM-10) (Figure 2) [106,107].

AGE-flRAGE binding initiates a number of signalling cascades (Figure 2), including mitogen-activated protein kinase (MAPK) [108,109,110], janus kinase/signal transducer and activator of transcription (JAK/STAT) [111,112] and rho GTPases [113,114,115]. Furthermore, RAGE ligand binding results in nuclear translocation of transcriptional factors such as nuclear factor kappa B (NF-κB) [94,116] and early growth response protein 1 (Egr-1) [117]. These pathways, activate downstream pro-inflammatory pathways inducing chemotactic cellular migration, proliferation and apoptosis (Figure 2) [118]. These are in line with the known physiological role for RAGE in host-pathogen defence.

Animal and human data indicate that RAGE expression is likely modulated by environmental factors, including AGEs [119,120] and disease states such as diabetes [98,101]. Whilst increased levels of full length, membrane-bound RAGE are associated with disease progression and poor outcomes in both animal models and humans [100,101,104], circulating soluble forms are inversely correlated with metabolic parameters, including body mass index (BMI), serum triglycerides, HbA1c, insulin resistance [121], and chronic disease [122,123,124,125,126,127], and positively associate with longevity [128,129]. This is supported by animal studies that have conclusively shown that administration of sRAGE or esRAGE is protective or has positive outcomes for a number of disease states [101,130,131,132]. This evidence has led to the hypothesis that soluble forms of RAGE may act as a competitive antagonist for AGEs, modulating their ability to interact with receptors such as membrane-bound RAGE, thereby preventing downstream signalling (Figure 2) [133,134].

### 4.2. AGER1

AGER1, also known as OST48, is a 48-kDa, type I transmembrane receptor protein [135] that localises to the plasma membrane [136] and endoplasmic reticulum (ER) [137]. The primary role of OST48 is as a subunit of the multiprotein oligosaccharyltransferase complex responsible for the N-linked glycosylation of asparagine residues during protein translation at the ER; therefore, it is expressed in all tissues, but particularly glandular cells (proteinatlas.org) [138,139]. In AGE biology, it is postulated to facilitate AGE clearance by lowering AGE levels in the intra- and extra-cellular environment and facilitating their clearance into the urine [140,141]. Additionally, it appears to be a negative regulator of the inflammatory response in some cell types [142].

In several chronic disease states, such as diabetes, CKD and autoimmune diseases AGER1 levels are down-regulated [82]. Dietary AGEs and other dietary factors appear to influence AGER1 levels. For example, restricting of AGE consumption has been shown to increase AGER1 in peripheral blood mononuclear cells (PBMCs) in CKD patients [140] and in the kidney, spleen and liver of healthy, but aged, mice [143]. Similar increases in PBMC AGER1 have been observed in clinical trials of a Mediterranean diet, low in AGEs [144,145] and a diet high in monounsaturated fatty acids (PUFAs) [146]. However, global over-expression of AGER1 increased urinary AGE clearance and improved insulin effectiveness in experimental diabetes in mice, but resulted in increased tubulointerstitial fibrosis [147]. Similarly, overexpression of AGER1 in the podocytes of mice resulted in glomerulosclerosis and podocyte damage and a decline in GFR, despite increasing renal AGE clearance, this was further exacerbated by diabetes [148].

## 5. AGE-Mediated Pathology

### 5.1. AGEs Can Induce Both Receptor Mediated and Non-Receptor Mediated Pathology

AGE-mediated pathology can result from the deposition of AGEs and the subsequent crosslinking of structural proteins in cells and organs and tissues or through AGE-receptor interactions, both of which impair tissue and cellular function. For example, collagen AGE modification and crosslinking of other extracellular matrix proteins leads to structural alterations, including changes in packing density [149] and surface charges [150] and the loss of structural integrity. This results in stiffening of the vasculature and expansion of cellular basement membranes in diabetes and CKD. AGE-RAGE binding activates pro-inflammatory signal transduction cascades increasing cytokine and growth factor expression [19,151,152]. As such, the AGE-RAGE axis is implicated in diabetes complications, CKD and end-stage renal diseases (ESRD), but also many other chronic diseases, including Alzheimer’s disease, atherosclerosis, cataracts, Parkinson’s disease, sarcopenia, vascular dementia and aging [153,154]. In humans, several studies show strong associations between circulating AGEs and inflammatory markers in elderly [155], young (adolescent) [155], and diabetic [156] populations. However, low and high AGE dietary studies in humans have been contradictory in their findings with some authors reporting no effects of dietary AGEs on systemic inflammation [157].

### 5.2. Dietary AGEs and AGE Pathology

The contribution of dietary AGEs to various pathologies remains to be fully elucidated. In rodent studies, chronic dietary exposure to excess CML results in damage to the glomerulus of the kidney, and albumin in the urine [158], insulin resistance [159,160], and insulin secretory defects [161,162]. In both wild type and T2D models, excess AGE dietary consumption elevates fasting plasma glucose levels, and worsens proteinuria, albuminuria and kidney [79,163,164,165] and liver [85] injury. Conversely, low AGE diets extend lifespan and improve age-related glucose abnormalities and renal outcomes and increase AGER1 expression in C57Bl6K mice [143]. Low AGE diets can also prevent type 1 diabetes (T1D) when administered to pregnant and weaning mothers in non-obese diabetic (NOD) mouse models of T1D [166] and attenuate insulin resistance and improve vascular and renal outcomes [81,167,168] and wound healing [169] in other diabetic models.

In humans, many benefits have been shown with AGE restriction in the diet. In individuals with T2D and renal failure, excessive AGE intake positively correlated with serum biomarkers of oxidative stress, inflammation, endothelial dysfunction, hyperglycaemia and hyperlipidaemia [170,171]. Conversely, dietary AGE restriction in healthy individuals and those with T2D have demonstrated favourable outcomes in circulating 8-isoprostanes and tumor necrosis factor alpha (TNFα) [140,141], improvements in cognitive function [172] and insulin sensitivity [141,173]. The effects of dietary AGEs on insulin sensitivity is vitally important, since abnormalities in glucose homeostasis are potent risk factors for CKD development and progression. In animal models, increasing circulating AGEs [174,175] and consumption of diets high in AGEs [160,162,167,176] irrefutably result in decreased insulin sensitivity, independent of other dietary factors [119,160]. In humans, several studies have demonstrated a relationship between high AGE diets and insulin signalling defects [177] with acute changes in insulin secretion following high AGE meal challenges [178,179]. Additionally, two independent randomised crossover dietary intervention studies have found that low AGE diets improve insulin sensitivity [173,180], and renal function [79] following a 2-week, or 4-week, high and low AGE dietary protocol, respectively. Although a recent study failed to recapitulate these findings with regard to insulin sensitivity [157].

### 5.3. Dietary AGEs and the Microbiome

One of the most interesting frontiers in AGE biology is their interaction with the microbiome. Many dietary AGEs are not easily absorbed by the small intestine, passing instead to the colon where they are available for metabolism by the colonic microbiome [53]. Only limited in vivo data exists regarding the effects of ingested AGEs on the colonic microbiome. In adolescent males, CML intake was negatively associated with *Lactobacilli* and positively associated with Enterobacteria following two weeks on a high or low AGE diet (randomised crossover design) [181]. Meanwhile, in dialysis patients, dietary AGE restriction resulted in alterations to the gut microbiome [182]. However, both studies are limited by small study size and understanding the physiological relevance of reported microbial changes is challenging. In mice however, 22 weeks consuming a diet enriched with the AGE MG-H1 induced significant microbial changes in the gut that were associated with metabolic dysregulation and increased systemic inflammation [183]. Similarly, in mice fed a heat-treated high AGE diet for 24 weeks, significant expansion of *Helicobacteraceae* and contraction of *Saccharibacteria* populations was observed, which coincided with an increase in gut permeability, increased circulating lipopolysaccharide (LPS), complement activation and onset of changes associated with early CKD. This was attenuated by a diet high in resistant starch [165].

It is reasonable to assume that probiotic or prebiotic supplementation that modulates the microbiome may, in turn, contribute to circulating and tissue AGE burden. However, whether this occurs through modulation of enteroendocrine factors, inflammation or gut-barrier permeability remains to be fully elucidated. Indeed, a recent meta-analysis found that pre-, pro- and symbiotic-supplementation reduces fasting insulin levels, hyperinsulinaemia and circulating AGEs in individuals with diabetes [184], suggesting the link between the microbiome, glucose homeostasis and AGE burden is an important one. However, these biota-based effects were modest, and the authors highlight the challenges of this type of human study where variation is significant and highly dependent on microbiome composition at baseline. A small, randomised control trial in women with T2D found supplementation with the prebiotic resistant dextrin significantly increased serum sRAGE and reduced the serum concentrations of AGEs CML, pentosidine, malondialdehyde (MDA) [185]. However, the study included only a small number of participants and did not assess microbiome composition. Further, the prebiotic supplementation with dextrin also reduced blood glucose concentrations, body weight and energy intake, and so it is almost impossible to determine the causative factors driving changes in circulating AGE concentrations. Another randomised crossover study comparing the effects of prebiotic supplementation on circulating AGEs, in individuals with pre-diabetes, may help further elucidate the contribution of the microbiome to AGE burden within the body [186]. This study has completed recruitment (ACTRN12613000130763) but to our knowledge the data have not been published. This remains an interesting and novel area requiring further investigation.

## 6. AGEs and Kidney Disease

Regardless of the presence of diabetes, previous studies have established a link between elevated circulating AGEs and a progressive decline in renal function [20,21,75,187,188,189]. Since dialysis does not as effectively remove AGEs [190], individuals receiving haemo- or peritoneal-dialysis show higher circulating AGE concentrations compared to healthy individuals [191,192]. Circulating AGEs also show a positive association with markers of inflammation and oxidative stress in uremic patients [193] and predict cardiovascular disease mortality in stable renal transplant recipients [194]. Moreover, elevated circulating concentrations of sRAGE are positively associated with the incidence of diabetic kidney disease (DKD) [195] and CKD [196,197]. In the ADVANCE study of 3763 individuals with T2D, both circulating AGEs and sRAGE were positively associated with incident CKD, progressive CKD, and mortality which led researchers to propose that the AGE:RAGE axis may represent an important target for the prevention and management of diabetic nephropathy [198].

The relationship between endogenous AGE production and renal function has been demonstrated in studies using skin collagen fluorescence as a biomarker of AGE burden. Skin collagen autofluorescence is significantly increased in CKD patients without diabetes [199,200,201] and predicts those individuals with progressive CKD [199,202]. In individuals with diabetes and established CKD, skin autofluorescence is also inversely associated with estimated glomerular filtration rate eGFR, a measure of renal function [203,204], and positively associated with incidence of CKD [204] and mortality [205]. Furthermore, in young adults with T1D but without established kidney disease, skin autofluorescence and eGFR predicted approximately 25% of variance for DKD risk [206]. In a longitudinal study of individuals with T1D, the Diabetes Control and Complications Trial (DCCT), and its follow-up study, Epidemiology of Diabetes Interventions and Complications (EDIC), skin collagen CML concentrations predicted individuals who developed nephropathy and cardiovascular disease [207,208], and this was independent of glycaemic control. These results indicate that AGE production and its accumulation is implicated in CKD as more than simply a biomarker for glucose homeostasis.

Increases in AGE concentrations in individuals with CKD are likely due to both reduced renal clearance and accelerated endogenous AGE production. In vitro, AGEs (CML and pentosidine) and reactive precursors form more rapidly and to significantly greater levels in serum from uremic patients compared to healthy controls [209]. This suggests that circulating factors that promote or stimulate the rapid formation of AGEs are present in the bloodstream of individuals with reduced renal function. In diabetes, increased circulating and urinary excretion of AGEs appears to predict development and progression of DKD [210], even in young adult populations without established kidney disease [211]. However, in a prospective study of T1D patients with early DKD, there was a decrease in circulating AGE concentrations associated with increased urinary CML clearance. This was associated with future rapid GFR decline [212]. A decrease in circulating AGEs might be the result of glomerular hyperfiltration, which is common in diabetes and may lead to a reduction in low molecular weight AGEs [213]. However, in cross-sectional data from patients with diabetes of long duration, urinary AGEs were positively associated with albuminuria, independent of isotopic GFR [210], suggesting tubular processing and excretion of AGEs is an important determinant of urinary AGE concentrations, as suggested by Kern et al., who also concluded that urinary AGEs are a useful early marker of tubular damage [214]

### 6.1. Mechanisms by Which AGEs Damage the Kidney

AGEs accumulate in the renal compartment where they mediate kidney damage (Figure 3). Cross-linking of matrix proteins by AGEs leads to stiffness and altered structural function at sites such as the glomerulus, peritubular vasculature and arterioles of the kidney, promoting glomerulosclerosis, atherosclerosis and thickening of the basement membrane [26]. AGE accumulation in the glomerulus is also associated with podocyte epithelial mesenchymal transition [215,216]. Similarly, in vitro exposure to high concentrations of AGEs induces tubular-epithelial-myofibroblast transition via RAGE dependent pathways, contributing to tubulointerstitial fibrosis [217,218,219]. AGE interactions with membrane-bound forms of RAGE, ultimately lead to the induction of a number of pro-inflammatory cytokines and chemo-attractants [220,221,222,223,224,225] via the activation of Nuclear Factor kappa B (NF-κB) and JAK-STAT pathways, promotion of inflammation via interleukin 1 β (IL-1β) and TNFα pathways [226,227], and the stimulation and production of NADPH oxidase and mitochondrial derived ROS [101,228]. These aforementioned pathways are well described in the development of fibrosis, glomerulosclerosis, apoptosis and cell death and are characterised contributors to the progression of DKD and CKD in both humans and pre-clinical models [19]. Accumulation of RAGE ligands, including AGEs, stimulates increased RAGE expression on podocytes in humans and rodent models [98,100,101] and numerous studies have shown RAGE knock-out in diabetic mice improves renal injury [228,229,230]. 

### 6.2. Renal Handling of AGEs

In vivo uptake and trafficking studies in healthy rats have shown that intravenously injected low molecular weight AGE-peptides, but not AGE-BSA, are freely filtered at the glomerulus and detected in urine [74]. However, both in vivo and in vitro data suggest that, once filtered, AGEs bind to and/or enter proximal tubule cells [63,74,217,231,232], where their removal is dependent on lysosomal degradation and autophagy [233]. This is supported by in vitro studies in renal proximal tubule cells, which can degrade AGE-modified albumin and release them into the supernatant However, exocytosis of glycated albumin peptide fragments was slow compared to non-glycated fragments, suggesting that renal tubule processing of AGE modified proteins is impeded by glycation [231]. Ultimately, this may lead to increased accumulation of AGEs within proximal tubule cells and disrupt autophagosome lysosomal pathways [234]. Supporting this, Tessier et al. showed that in mice, the kidneys were amongst those organs with the highest levels of dietary AGE accumulation following 30 days on an 13C-CML-BSA enriched diet [69]. Overall, it appears that the transport of free- and/or protein-bound AGEs across the renal filtration barrier and tubular cells may depend on protein size and the degree and site of glycation.

### 6.3. Exogenous AGEs and Kidney Function

There is a paucity of studies examining the longitudinal effects of high AGE consumption on risk of kidney disease in healthy humans. One prospective study using dietary questionnaires demonstrated a 2-fold increase in CKD risk with high AGE consumption from dietary fat sources [235] even after adjustment for diabetes and hypertension, however, human studies examining the effects of dietary AGEs on direct measures of kidney function are lacking [236]. A small pilot study of 10 healthy participants demonstrated that the degree of glycation of isocaloric protein loads differentially affected renal haemodynamics, with high AGE loads increasing renal perfusion and renal oxygen consumption [237]. In overweight individuals without diabetes, a randomised cross-over study showed that increased AGE consumption for 2 weeks increased systemic inflammation, albuminuria and decreased eGFR, while paradoxically increasing urinary AGE clearance and reducing circulating levels [79]. In agreement, several studies in patients with diabetes [156] and healthy individuals [173] have demonstrated a significant effect of increased dietary AGEs on other determinants of kidney disease such as increased endothelial dysfunction and oxidative stress [156], as well as adverse effects on insulin sensitivity and circulating lipids [173] when compared to low AGE diets. Even a single oral AGE challenge was sufficient to induce endothelial dysfunction and oxidative stress in healthy individuals and in those with diabetes [238]. However, not all studies of high/low AGE dietary interventions in healthy adults have reported changes in endothelial dysfunction and inflammation [239,240]. This is possibly because the AGE content of the high AGE diets used were similar to that of a typical Western diet and may have been comparable to the patients’ diets at baseline.

In animal models, significantly more work has been undertaken to measure the direct effects of exogenous AGEs on the kidney. Animals fed a high CML diet accumulate AGEs preferentially in the kidneys [69,241] and, in models with established kidney disease, such as the remnant kidney model, high AGE diets for a period of 5–13 weeks (study dependent) increased proteinuria [164,242,243], fibrosis and glomerular injury [243]. In healthy rats without kidney injury, a 4-week high fat/high AGE diet increased serum creatinine, suggestive of decreased GFR, which was associated with enhanced kidney CML deposition and markers of oxidative stress and inflammation [244]. In contrast, a more recently published study examining the effects of a high CML diet for 18 months in healthy male mice reported no effects on kidney ageing [245]. This may indicate that where there is no pre-existing kidney injury, a genetic predisposition to kidney disease, or risk factors such as insulin resistance, enrichment of the diet with single AGEs is not a potent driver of renal changes compared to a baked heterogeneous AGE diet.

While the majority of studies have reported on the impact of AGEs on kidney disease risk factors (e.g., diabetes, hypertension), few have assessed effects on kidney function. However, as highlighted above, there is much to be untangled regarding whether kidney changes (functional and structural) are actioned via direct renal uptake and trafficking of ingested AGEs or mediated indirectly by AGE effects on, and renal interactions with, the vascular, GI tract, enteroendocrine systems and the microbiome.

### 6.4. Therapeutic Targeting of AGEs in Kidney Disease

There are several classes of therapeutic agents aimed at targeting AGEs. A thorough review of these can be found here [246]. Several of these agents have been trialled in experimental models of DKD or CKD and in clinical trials.

#### 6.4.1. Carbonyl Scavengers

Carbonyl scavenging is one approach that has been trialled quite extensively since the early 1990s. Aminoguanidine (Pimagedine), a carbonyl scavenger, was found to lower AGEs and attenuate the DKD-driven rise in albuminuria and prevent mesangial matrix expansion in rats [247]. However, aminoguanidine was found to bind to a number of targets including functional endogenous carbonyls, making it inappropriate as a therapeutic agent [248]. Furthermore, two clinical trials were performed, ACTION I and ACTION II, but they failed to show efficacy and ACTION II was terminated early due to safety concerns and off target effects [248].

#### 6.4.2. Vitamin B and Its Derivatives

The B group vitamins and a number of their derivatives have also been trialled as AGE lowering therapies. These include thiamine, benfotiamine, and pyridoxamine [246]. These agents are thought to reduce AGEs by increasing the activity of the thiamine-dependent enzyme, transketolase. The resulting stimulation of the pentose phosphate pathway should, in theory, reduce glycolytic intermediates. Clinical trials of thiamine and benfotiamine and their efficacy in the context of kidney disease are lacking. However, pyridoxamine, which is thought to act by inhibiting the conversion of Amadori products and by chelation of dicationic metal ions has shown efficacy in animal models of kidney disease [249,250,251] and to effectively reduce CML, CEL and transforming growth factor β (TGFβ) in DKD patients [252]. Unfortunately, a phase III clinical trial examining the efficacy of pyridoxamine in DKD, PIONEERIII, began recruiting in 2014 but was terminated due to financial constraints in 2016. As yet, the study has not been recommenced.

#### 6.4.3. AGE Cross Link Breakers

N-phenacylthiazolium bromide (PTB) was the first reported compound capable of breaking the crosslinks formed by AGEs, leading to the development of a class of compounds known as “cross link breakers” [253]. While PTB demonstrated efficacy for reducing AGEs in diabetic rodents [254], PTB was not effective at preventing or attenuating DKD in rodents [255,256] and due to its in vivo instability further compounds were developed [246]. 4,5-dimethyl-3-phenacylthiazolium chloride (ALT-711^®^, Alagebrium chloride) is the most well studied of the PTB analogues. Alagebrium has cross link breaking properties but also appears to be a MG scavenger, and exhibit antioxidant and chelating properties. It has demonstrated efficacy in reducing blood pressure, mesangial matrix expansion and tubulointerstitial fibrosis in rodent models of DKD [257,258] and hypertension [259]. Due to loss of the clinical sponsor and patent during the global financial crisis, various Phase II clinical trials both in heart failure (NCT00739687 and NCT00516646) and DKD (NCT00557518) had to be prematurely terminated.

#### 6.4.4. RAGE Blockade

AGE-RAGE blockade is another potential therapeutic target for chronic kidney disease. RAGE knockout in mice prevents or attenuates DKD [101,168,260] and deletion of RAGE from bone marrow derived cells reduced renal functional changes as well as immune infiltration seen with experimental (STZ induced) diabetes [261]. Long term treatment with the RAGE decoy receptor, sRAGE, has been shown to improve measures of DKD in the db/db model of T2D and obesity [101] and RAGE blockade by antibodies was effective at slowing progression of DKD in models of T1D [262] and T2D [263]. More recently, AGE targeted aptamers that prevented AGE-RAGE signalling protected against DKD in a mouse model of T2D and obesity [264]. Similarly, RAGE aptamers given early in diabetes or at diabetes induction showed significant potential attenuating progression of DKD in rats [265] and showed effectiveness in reducing markers of kidney injury in uni-nephrectomised deoxycorticosterone acetate (DOCA)/salt-induced hypertensive mice [266]. Although there is considerable pre-clinical evidence for RAGE blockade as a therapeutic option in kidney disease, to our knowledge no agents have reached the clinical trial phase for kidney disease.

## 7. Conclusions

AGEs are an important mediator of pathology, particularly in diabetes and kidney disease where AGE homeostasis is unbalanced by impaired clearance and increased endogenous production. Given the evidence presented, dietary AGEs appear to be an important contributor the body’s AGE pool, but may also act to accelerate endogenous AGE production through increased oxidative stress, endothelial dysfunction and by precipitating glucose abnormalities. Critical experiments need to be performed in humans to understand the extent to which dietary AGEs contribute to the onset and progression of CKD in humans, since AGE-lowering strategies show some promise in clinical studies performed to date.

## Figures and Tables

**Figure 1 nutrients-14-02675-f001:**
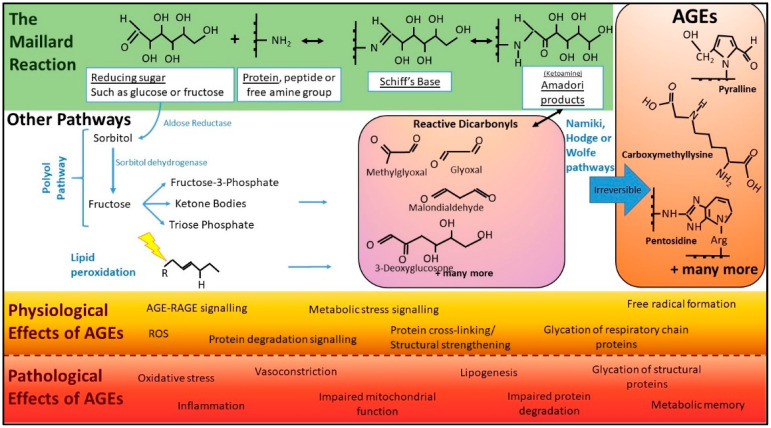
Basic pathways of AGE chemistry and associated physiology and pathology. In the classical Maillard reaction (green), AGEs are formed when reducing sugars and proteins interact. Initially, reaction intermediates termed Schiff bases are formed. These subsequently rearrange into ketoamines known as Amadori products or early glycation products. These are more stable products than Schiff bases but, at this point in the reaction, the chemistry is still reversible. After further chemistry, the reaction becomes irreversible and forms AGEs (orange). There are several other cellular pathways, including the polyol pathway (shown) and lipid peroxidation (where oxidants, such as free radicals, “attack” (lightning bolt) lipids that contain carbon-carbon double bonds), that can lead to the formation of reactive dicarbonyls, “termed carbonyl stress” (purple) rapidly accelerating the formation of AGEs. Beyond proteins, lipids and DNA can also undergo glycation, leading to AGEs. Pathways such as the polyol pathway only occur inside the body. AGEs have both physiological roles (orange) and pathological effects (red). (Abbreviations: AGEs, advanced glycation end products; ROS, reactive oxygen species).

**Figure 2 nutrients-14-02675-f002:**
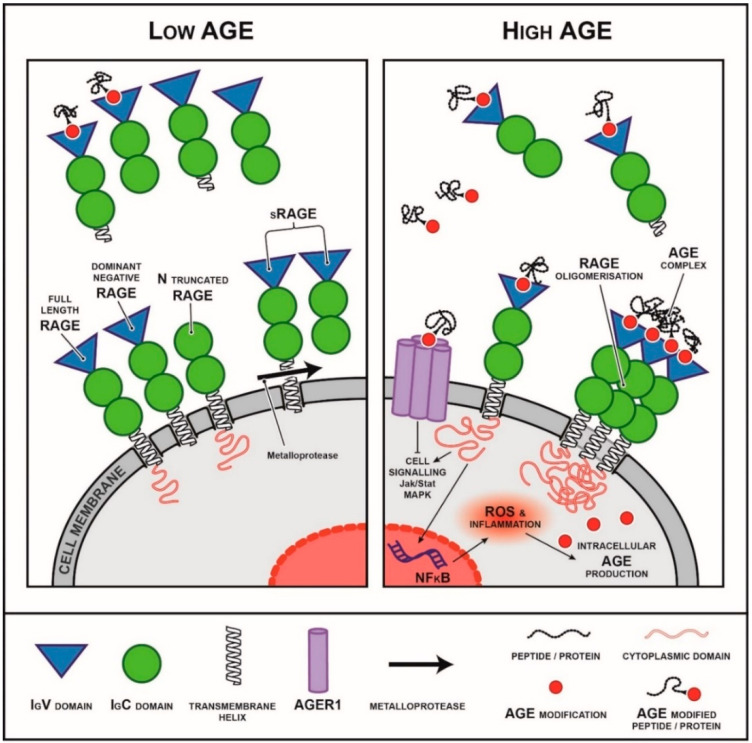
Isoforms of the receptor for AGEs (RAGE) and their interactions with AGEs and downstream pathways activated. RAGE is a multi-ligand member of the immunoglobulin superfamily of receptors. It can exist as different isoforms including membranous isoforms (full length RAGE, dominant negative RAGE, N truncated RAGE) as well as soluble secreted and cleaved isoforms. The cytoplasmic domain is essential for RAGE signalling. The secreted isoforms are thought to act as decoys, binding to RAGE ligands and preventing them from binding to membranous forms of RAGE activating downstream signalling. In a low AGE environment (left), circulating sRAGE is believed to act as a decoy receptor, binding circulating RAGE ligand such as AGEs and preventing binding to membranous isoforms of RAGE. In a high AGE environment (right), circulating sRAGE levels are commonly decreased or sRAGE capacity is saturated and is no longer sufficient to prevent RAGE downstream signalling.

**Figure 3 nutrients-14-02675-f003:**
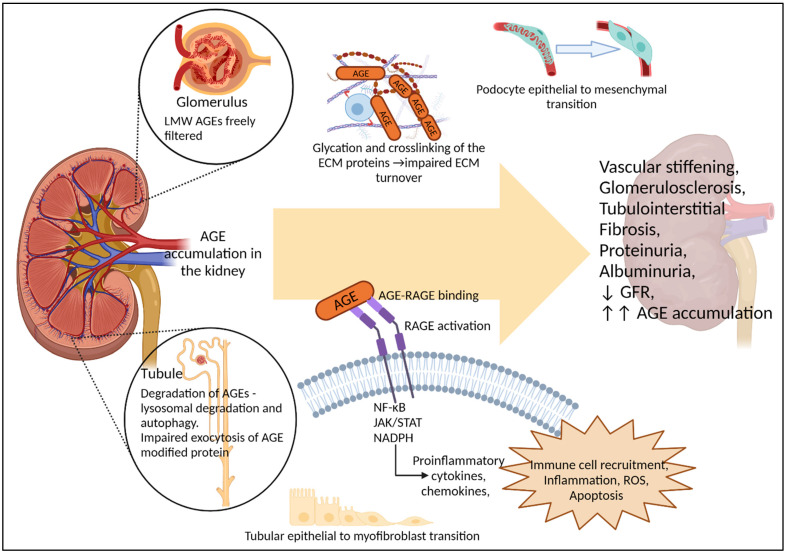
Renal handling of AGEs and their contribution to renal pathology. In vivo evidence from rodents suggests AGEs accumulate in the kidney, which is also a major site for AGE clearance. Low molecular weight (LMW) AGEs are freely filtered by the glomeruli, while lysosomal degradation and autophagy of AGEs appears to occur within the tubules, with AGE modification impairing exocytosis of proteins by the tubular cells. AGE accumulation in the kidney contributes to a number of pathological pathways, including glycation and crosslinking of structural proteins, dedifferentiation of specialised epithelial cells such as podocytes, and RAGE activation leading to further inflammation, ROS and cellular apoptosis. Together these can contribute to, or exacerbate hemodynamic changes, glomerulosclerosis, tubulointerstitial fibrosis, proteinuria, albuminuria and loss of GFR. As renal function declines, renal capacity to excrete AGEs is reduced, leading to increased AGE burden within the body, as is seen with CKD. However, evidence that AGEs alone, in the absence of diabetes or underlying renal conditions, can induce renal dysfunction has largely come from rodent models. with only limited associative studies in humans.

**Table 1 nutrients-14-02675-t001:** Uptake, elimination and biodistribution studies of AGEs and their precursor Amadori products.

**Author**	**Amadori Product (AGE Precursor)**	**Study Population**	**Methodology**	**Key Findings**	**Measurement Technique**
[58] Erbesdobler and Faist, 2001	Fructoselysine, Fructoseglycine	Rats and Humans	In vitro, everted gut sack, in vivo ligated jejunal segments	1–3% of ingested Amadori products were detected in urine Faecal output persisted for several days	^14^C Radioactivity
[59] Forster et al. 2005	Pyralline, fructoselysine, pentosidine	Humans	Exclusion diet, followed by a high AGE test meal	Low AGE diet lowered urinary pyralline and fructoselysine by 90%, and pentosidine by 40% Post high AGE test meals, 50% of ingested pyralline and 60% of pentosidine were recovered in urine. 2% of peptide bound pentosidine was recovered	Reversed Phase HPLC with UV detection
[60] Hultsch et al. 2006	Fructoselysine	Rats	Injected and gavaged ^18^F flourobenzoylated fructoselysine. Biodistribution and catabolism study performed using PET scanning	Orally ingested fructoselysine did not appear in circulation or tissues Injected fructoselysine rapidly appeared in circulation and cleared into urine over 60 min	PET scanning/Radioactivity counting
[61] Schwenger et al. 2006	Lactuloselysine	Humans	Diet administered to healthy, diabetic and renal failure patients. Plasma concentrations and cumulative urinary excretion examined	Only small acute increase (2% of administered dose) appeared in urine, and plasma concentrations did not change	Reverse Phase HPLC
**Author**	**AGEs**	**Study Population**	**Methodology**	**Key Findings**	**Measurement Technique**
[62] Liardon et al. 1987	CML	Rats	Diets varying in quantity AGEs- timed urine collection and analysis for CML	Urinary CML varied according to diet suggesting it came from exogenous sources	Mass Spec
[55] Koschinsky et al. 1997	Protein bound AGEs	Humans (incl. diabetes with/without DKD)	Single meal challenge, AGE egg white or fructose + egg white	Absorption estimated to be ~10% of total AGEs ingested Renal excretion was ~30% of total absorbed In DM patients, excretion inversely correlated with albuminuria and was slower	ELISA
[63] Miyata et al.1998	Pentosidine	Rats	IV injection with radiolabelled pentosidine, urine collected over 72 h	Radioactivity peaked 1 h after administration 80% of radioactivity recovered after 72 h, only 20% was intact pentosidine Pentosidine reabsorption by proximal tubule after glomerular filtration	Radioactivity/ ELISA
[54] He et al. 1999	AGE-Ovalbumin	Rats	Fed a single dose of ^14^C or ^125^I labelled AGE-ovalbumin. Collected tissues, plasma and 72 h urine collection.	~10% of radiolabelled AGEs absorbed. Many AGEs still bound to peptides	Radioactivity/ ELISA
[64] Bergman et al. 2001	CML and CEL (free)	Rats	Biodistribution study of intravenously administered ^18^F labelled AGEs using PET scanning and radioactive counting	CML and CEL accumulated in liver and kidney at 20 min, were rapidly excreted into urine and undetectable by 2 h. Small amounts also accumulated in spleen, pancreas, heart, lungs stomach and intestine Estimated renal clearance of CML and CEL was 1.73 mL/min and 3.09 mL/min, respectively	PET scanning
[65] Somoza et al. 2006	CML, LAL, FL (Casein linked)	Rats	Casein linked AGE feeding (2 dosages) to metabolic caged rodents	AGE-modified casein was absorbed less CML modified casein demonstrated highest recovery in urine and faeces Only CML impacted kidney and liver weight and urinary excretion of AGEs The high CML diet increased plasma CML 5-fold	HP-LC-UV fluorescence
[57] Sebekova et al. 2008	CML	Human infants	Comparison of circulating AGEs between breast and formula fed infants	Plasma CML was 60% higher in formula fed infants than breast milk fed infants	LC-MS/MS
[66] Roncero-Ramos et al. 2013	CML	Rats	88 days on high or low AGE diet—Bread crust or it’s insoluble (HMW) or soluble fractions (LMW)	Circulating CML did not differ between groups Urinary CML did not correlate with diet Faecal excretion was influenced by diet Increased CML in cardiac tissue and tail tendon	HPLC-MS/MS
[67] Alamir et al. 2013	Extruded or non-extruded protein diet CML	Rats	6 weeks feeding on extruded or non-extruded protein diet or single oral free CML challenge	Protein bound serum CML levels increased 4-fold after single oral dose and remained high for 4 h of monitoring Unbound CML remained unchanged	LC-ESI-MS/MS
[68] Xu et al. 2013	CML (free)	Mice	Biodistribution and elimination study ^18^F labelled CML in mice. Tracer labelled CML was administered either IV or intra-gastrically	IV administered CML quickly distributed in bloodstream and cleared via kidneys within 20 min CML detected in a number of other organs Intragastric administered CML showed minimal absorption	PET scanning
[69] Tessier et al. 2016	CML (protein bound)	Mice	30 days feeding with a diet enriched with ^13^C-CML that could be differentiated from native CML in C57BL/6J and RAGE knock out mice	Mice showed accumulation of dietary derived CML in all tissues analysed, but highest in kidney, intestine and lungs and independent of RAGE	Stable isotope dilution analysis LC-MS/MS
[70] Tsutsui et al. 2016	AGE-Albumin	Mice	Single IV injection of Cy 7.5 labelled AGE-BSA. Fluorescence kinetics assay performed	Injected AGE-BSA showed strong localisation to liver and impaired clearance compared to BSA AGE-BSA co-localised with scavenger cells of liver	IVIS whole animal in vivo imaging system

Abbreviations: BSA, Bovine Serum Albumin; CML, Carboxymethyllysine, CEL, Carboxyethyllysine; DKD, Diabetic Kidney Disease; DM, Diabetes Mellitus; ELISA, Enzyme Linked Immunosorbent Assay; FL, Fructoselysine; HMW, High Molecular Weight; HPLC, High Performance Liquid Chromatography; IV, Intravenous; IVIS, In Vivo Imaging System; LAL, Lysinoalanine; LC- MS/MS, Liquid Chromatography Mass Spectrometry; LC-ESI-MS, Liquid Chromatography-Electrospray Ionization Mass Spectrometry; LMW, Low Molecular Weight; PET, Positron Emission Tomography.

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
