# Peer review of "Advanced Glycation End Products (AGEs) and Chronic Kidney Disease: Does the Modern Diet AGE the Kidney?"

_nutrients, 2022, doi:10.3390/nu14132675_

Round 1
Reviewer 1 Report
This is a well written and well organised review of AGE in CKD. It describes the importance of AGEs, the biology of AGEs, and the clinical implications of AGEs and their handling and metabolism.
It raises and discusses important next questions to be addressed in research.
Author Response
We would like to thank reviewer 1 for their kind words in praise of our manuscript which required no further editing of the manuscript.
Reviewer 2 Report
The manuscript by Dr. Amelia K Fotheringham et al. reviews the mechanism involved in AGE formation and their regulation. Further the review gives an account on the contribution of AGE on kidney diseases. The review intends to analyze the contribution of dietary AGE in the pathogenesis in diabetic and non-diabetic kidney disease. The title is interesting however there should be clarity on the content.
While there are details on the sources of endogenous AGE production there is no notable information regarding the heterogeneity of the exogenous AGEs.
The title seems to be interesting however it seems to be misleading from the content provided in the manuscript. The review provides more emphasize on the AGE chemistry and the endogenous AGE production, their absorption and clearance but the pathological aspects of exogenous AGE-on aging-related abnormalities on glucose metabolites, renal out comes would have been more suitable for the title provided.
A table providing details of kidney specific conditions associated with AGE will be better.
The details provided are from a mix of animal and human studies, the details derived from animal studies shall be clearly mentioned.
Author Response
We would like to thank reviewer 2 for their comments and feedback. Please find below an itemised list of the comments and our response.
- While there are details on the sources of endogenous AGE production there is no notable information regarding the heterogeneity of the exogenous AGEs. Response: We would like to politely disagree with reviewer 2 and would like to draw attention to lines 103-105 (highlighted) and lines 187-188 (highlighted) where we specifically highlight the diversity of exogenous AGEs produced in cooking and food manufacturing. Indeed, the paragraph from line 95-105 is entirely focused on the fact that the Maillard chemistry can diverge generating a huge number possible AGEs. We have added some extra detail to this paragraph to better describe the diversity of AGEs found exogenously (lines 105-108). However, we would be happy to amend this paragraph further if reviewer 2 would like to clarify the type of specific detail that they would like to see included.
- The title seems to be interesting however it seems to be misleading from the content provided in the manuscript. The review provides more emphasize on the AGE chemistry and the endogenous AGE production, their absorption and clearance but the pathological aspects of exogenous AGE-on aging-related abnormalities on glucose metabolites, renal out comes would have been more suitable for the title provided. Response: We would like to politely disagree with reviewer 2 with regard to the title and it not accurately describing the content of the review. Our review discusses both the impact of AGEs consumed from the diet and the metabolic abnormalities precipitated by modern dietary habits which can in turn lead to excess endogenous AGE generation within the body. Further, we discuss how this increasing burden of AGEs may disproportionately affect the kidneys, as the organs largely responsible for AGE clearance. Therefore, in choosing a title for this review we wished to keep the title general to encompass both dietary derived and endogenously derived AGEs and their connection to CKD. We feel that the title Advanced Glycation End Products (AGEs) and chronic kidney disease: Does the modern diet AGE the kidney? neatly encapsulates all these factors. Furthermore, we believe the title is “catchy” and will generate interest. We would also like to point out that CKD is often likened to accelerated kidney aging with significant overlap between the pathogenesis of CKD and kidney aging. To increase clarity about the scope of the review, we have amended the abstract to better reflect the content of the manuscript, please see page 1, lines 17-18, 27 and 29 and line 42 where we have highlighted the similarity between CKD and kidney aging.
- A table providing details of kidney specific conditions associated with AGE will be better. Response: Chronic kidney disease refers to a very heterogenous group of diseases including diabetic kidney disease, focal segmental glomerulosclerosis, hypertensive kidney disease to name just a few. Given we are discussing not only the effects of dietary derived AGEs but also endogenously produced AGEs it would be very difficult to summarize this into a single table which could conceivably include more than 50 references. We believe we have concisely and clearly summarized the evidence linking AGEs (diet and endogenously derived) and renal function/CKD in the final third of the review and therefor a table would not add further value.
- The details provided are from a mix of animal and human studies, the details derived from animal studies shall be clearly mentioned. Response: We have endeavoured to be very clear in highlighting where evidence comes from human or animal sources. Table 1, for example, is very clear regarding the models used for uptake and biodistribution studies and we have highlighted in green many instances in the main text where we have tried to keep this clear. However, we have added further detail to the manuscript to improve the clarity where appropriate. Please refer to the changes in red on lines 191, 209, 213, 224, 277, 281, 326, 504, 520, 595, 638.
Reviewer 3 Report
This review discusses the pathways that drive AGE formation and regulation within the body along with a focus on the contribution of dietary AGE consumption to these processes. The authors also analyse the contribution of AGEs to kidney disease, the evidence for dietary AGEs in driving pathogenesis in diabetic and non-diabetic kidney disease and the potential for AGE targeted therapies in kidney disease. Here are some suggestions for further revisions.
1. Abstract: Ref. [1] shall be removed.
2. Please check all abbreviations, such as T2D.
3. Strongly suggest the authors depicting a whole picture to describe the current findings of AGE-mediated pathology. Also, a picture to present the relationship between AGEs and kidney disease is necessary.
Author Response
We would like to thank reviewer 3 for their comments and feedback. Please find below an itemised list of the comments and our response.
- Abstract: Ref. [1] shall be removed.
This has been done (see marked up manuscript)
- Please check all abbreviations, such as T2D.
Thank you, for bringing this to our attention we have checked all abbreviations to make sure the are appropriately defined the first time they are used. Please refer to lines 45, 245, 369, 415, 440, 496, 497, 611
- Strongly suggest the authors depicting a whole picture to describe the current findings of AGE-mediated pathology. Also, a picture to present the relationship between AGEs and kidney disease is necessary.
Thank you, for this suggestion, we have added a figure demonstrating the association between AGE mediated damage and renal function/kidney disease (see figure 3, page 17). However, we feel that figure 1 demonstrates the mechanisms of AGE mediated pathology succinctly in the context of the AGE chemistry and physiological role for AGEs. Therefor we believe that an additional figure describing AGE-mediated pathology would be repetitive especially given the addition of Figure 3 which also covers mechanisms of AGE pathology.